# Comparison of Ultrasound Scores in Differentiating between Benign and Malignant Adnexal Masses

**DOI:** 10.3390/diagnostics13071307

**Published:** 2023-03-30

**Authors:** Mar Pelayo, Irene Pelayo-Delgado, Javier Sancho-Sauco, Javier Sanchez-Zurdo, Leopoldo Abarca-Martinez, Virginia Corraliza-Galán, Carmen Martin-Gromaz, María Jesús Pablos-Antona, Julia Zurita-Calvo, Juan Luis Alcázar

**Affiliations:** 1Department of Radiology, Hospital HM Puerta del Sur, 28938 Móstoles, Spain; mar_pelayo@yahoo.es; 2Department of Radiology, Hospital HM Rivas, 28521 Madrid, Spain; 3Department of Obstetrics and Gynecology, Hospital Ramon y Cajal, 28034 Madrid, Spain; jsanchosauco@gmail.com (J.S.-S.); leohrcfetal@gmail.com (L.A.-M.); virginiacorraliza@gmail.com (V.C.-G.); cgromaz@yahoo.es (C.M.-G.); mjpablos@gmail.com (M.J.P.-A.); juliazuritacalvo@gmail.com (J.Z.-C.); 4Insight Technology Solutions, 28108 Madrid, Spain; javier.sanchez.zurdo@gmail.com; 5Department of Obstetrics and Gynecology, Clinica Universidad de Navarra, 31008 Pamplona, Spain

**Keywords:** transvaginal ultrasound, adnexal masses, IOTA simple rules, IOTA simple rules risk assessment, O-RADS, ADNEX model, CA125, diagnosis

## Abstract

Subjective ultrasound assessment by an expert examiner is meant to be the best option for the differentiation between benign and malignant adnexal masses. Different ultrasound scores can help in the classification, but whether one of them is significantly better than others is still a matter of debate. The main aim of this work is to compare the diagnostic performance of some of these scores in the evaluation of adnexal masses in the same set of patients. This is a retrospective study of a consecutive series of women diagnosed as having a persistent adnexal mass and managed surgically. Ultrasound characteristics were analyzed according to IOTA criteria. Masses were classified according to the subjective impression of the sonographer and other ultrasound scores (IOTA simple rules -SR-, IOTA simple rules risk assessment -SRRA-, O-RADS classification, and ADNEX model -with and without CA125 value-). A total of 122 women were included. Sixty-two women were postmenopausal (50.8%). Eighty-one women had a benign mass (66.4%), and 41 (33.6%) had a malignant tumor. The sensitivity of subjective assessment, IOTA SR, IOTA SRRA, and ADNEX model with or without CA125 and O-RADS was 87.8%, 66.7%, 78.1%, 95.1%, 87.8%, and 90.2%, respectively. The specificity for these approaches was 69.1%, 89.2%, 72.8%, 74.1%, 67.9%, and 60.5%, respectively. All methods with similar AUC (0.81, 0.78, 0.80, 0.88, 0.84, and 0.75, respectively). We concluded that IOTA SR, IOTA SRRA, and ADNEX models with or without CA125 and O-RADS can help in the differentiation of benign and malignant masses, and their performance is similar to the subjective assessment of an experienced sonographer.

## 1. Introduction

Ovarian cancer is one of the most lethal cancers of gynecological malignancies. Ovarian tumors are rare but with a high mortality rate and recurrence. Epithelial ovarian cancer remains the fifth cause of death in women and the first cause of death due to gynecological cancer [1]. Survival increases if treatment is performed at the initial stages in specialized referral centers with multidisciplinary teams, including gynecologist oncologists [2,3].

Although there is no universal screening program for ovarian cancer, ultrasound is now considered the technique of choice in the initial study of adnexal masses, especially using transvaginal ultrasound [4]. It has the advantage of being a harmless technique in which the patient does not receive radiation, is relatively economically affordable compared to other imaging techniques, such as computed tomography (CT) and magnetic resonance imaging (MRI), and is available to many gynecologists and radiologists worldwide [5]

The ultrasonographic classification of adnexal masses as benign or malignant is a challenge since it has a large subjective component, and the experience of the sonographer is relevant. It is also important to the ultrasound equipment used and its adjustments. To help in this task, in year 2000, the International Ovarian Tumor Analysis (IOTA) group defined the terms and definitions to describe tumor features that should be evaluated by ultrasound [6].

The subjective assessment of the sonographer has the greatest diagnostic validity in the characterization of the adnexal masses, and it’s completely dependent on the experience and training of the examiner [7,8,9]. However, the first contact with the patient is usually with a non-expert sonographer. In an attempt to make the classification of these lesions more objective and replace the expert’s experience as much as possible, different ultrasound scores have been proposed.

In 2008, the IOTA group published 10 rules (IOTA Simple Rules, SR) based on the descriptions proposed by Timmerman et al. to try to differentiate benign and malignant adnexal masses [10]. It contains ten ultrasound features to try to differentiate benign and malignant adnexal lesions. These are five descriptions suggestive of benignity (B features) and five of malignancy (M features). Study of malignancy, according to SR, considered three possibilities (If one or more malignancy characteristics appear and none of the benignity is present, the mass is classified as malignant; if one or more characteristics of benignity appear and none of the malignancy is present, the mass is classified as benign; if characteristics of both benignity and malignancy appear or no characteristic of malignancy or benignity appear, the mass is classified as “inconclusive”).

In 2016, the IOTA group published the so-called Simple Rules-Risk Assessment [11]. The parameters included are the same as the ones used in IOTA SR, adding the item “Oncology center”, defined as a tertiary referral center with a specific gynecological oncology unit. Benignity characteristics (B features) are defined as: Unilocular (B1), presence of solid component < 7 mm maximum diameter (B2), acoustic shadows (B3), regular multilocular tumor with maximum diameter < 100 mm (B4) and, negative color map (Score color 1) (B5). Malignancy characteristics (M features) are irregular solid tumor (M1), ascites (M2), 4 papillae (M3), multilocular irregular solid tumor with maximum diameter > 100 mm (M4), abundant color map (Score color 4) (M5).

In 2014, the Assessment of Different Neoplasias in the Adnexa (ADNEX) model was published [12]. This prediction model uses two clinical predictors: patient’s age (years), type of center (referral center defined as a tertiary referral center with a specific oncology unit vs. non-oncology center), one biochemical parameter (serum CA125, expressed in IU/mL) and six ultrasound values: maximum diameter of the lesion (mm), maximum diameter of the largest solid part (mm), more than 10 locules (yes/no), number of papillary projections (0, 1, 2, 3, >3), presence of acoustic shadows (yes/no), presence of ascites (yes/no). Results are provided as percentages of the risk of benignity, malignancy, or borderline for a lesion. ADNEX model has the advantage that it also provides the probability of malignancy in the different stages (stage I, II–IV, metastasis to the ovary) [13,14]. Note that CA125 would improve the differentiation between stage I and II–IV ovarian cancer.

In 2018, the American College of Radiology (ACR) defined the lexicon describing adnexal lesions based on IOTA descriptions [15], and in 2020 was published the Ovarian-Adnexal Reporting and Data System (O-RADS) [16]. It classifies adnexal masses according to 6 categories and includes probabilities or risk of malignancy and guidelines for management according to the risk category. O-RADS 0 means that the evaluation is incomplete; O-RADS 1 is used for normal ovaries or a physiological cyst with 0% probability of malignancy; O-RADS 2 (<1% malignancy) is set for an almost certainly benign lesion; O-RADS 3 is used for lesions with low risk of malignancy (1–9%); O-RADS 4 indicates lesions with an intermediate risk of malignancy (10–49%), whereas O-RADS 5 is associated with a high risk of malignancy (≥50%). To date, different flowcharts have been described to simplify its use [17].

The aim of the present study was to evaluate the different ultrasound scores (IOTA simple rules, Simple Rule Risk Assessment, O-RADS, ADNEX-Model with and without CA125) as well as the subjective impression of the experienced sonographer to determine how benign and malignant adnexal masses are differentiated.

## 2. Materials and Methods

### 2.1. Study Design

This is a retrospective study of a non-consecutive series of women diagnosed as having an adnexal mass who underwent surgery at the Department of Gynecology of a tertiary-care university hospital in Madrid (Spain) from January 2021 to December 2022. We performed 186 surgeries related to suspected adnexal masses. Two of them were excluded because they were finally diagnosed with cervical schwanoma and pediculated myoma. Other sixty-two lesions were studied by non-expert sonographers and were not included in the present study. The patient’s management was decided according to local clinical protocols and in a multidisciplinary tumor board session. Retrospectively, the clinical information and ultrasound images were reviewed. We obtained approval from the Local Ethics Committee.

### 2.2. Patients’ Inclusion and Exclusion Criteria

We included women with a definitive histological study of the adnexal lesion who previously had a gynecological ultrasound (maximum 180 days before) performed by an expert sonographer whose images were stored in the hospital’s PACs or in the ultrasound software. The criteria for surgical indication were those established by the clinic, personal history, or opinion of the center’s committee of onco-gynecologists. The histological study was carried out according to WHO criteria. Cases in which the ultrasound was not available or did not meet the appropriate quality criteria for its evaluation were not eligible.

### 2.3. Patients’ Data

We reviewed patients’ clinical records. Data collected from patients referred to their age, menopausal status, parity, body mass index (BMI), symptoms (asymptomatic, pain, gastro-intestinal symptoms, bleeding, and other), CA125 serum level (IU/mL), laterality of the lesion (right, left, bilateral), surgical procedure (laparoscopic/laparotomy, uni- or bilateral adnexal surgery, with or without hysterectomy), and final histopathology after tumor removal.

### 2.4. Image Capture and Analysis

Initially, all women underwent a transabdominal ultrasound for measurement of the tumor, especially in large masses. Then, the women underwent a transvaginal ultrasound. Patients who cannot tolerate the transvaginal approach (i.e., in virgo intacta) were examined transrectally to improve the visualization of details not detected by the abdominal route. Transvaginal or transrectal images were taken from a RIC 5-9D 4–9 MHz endovaginal probe and a RAB6-D 2–8 MHz transabdominal probe Voluson E8 (GE Healthcare, Ultrasound, Milwaukee, WI, USA) and Canon Aplio A (Canon Medical Systems corporation, Tokyo, Japan). As stated above, ultrasound examinations were performed by experienced gynecologists with more than 10 years of experience. All images were automatically stored in the ultrasound software and in the Picture Archiving and Communication System (PACS) immediately after the ultrasound scan. The images were reviewed by two ultrasound gynecological experts with more than 15 years of ultrasound analysis who were blind to the pathologic results and other image findings (CT and/or MRI). In case of disagreement, images were reviewed until a consensus was reached. The ultrasound characteristics of the adnexal lesions were classified using nomenclature and methodology proposed by the IOTA Group, including the application of Doppler color [5]. If the woman had more than one (bilateral) adnexal mass, the most complex mass was included, or if both were equal, the largest.

The ultrasound features assessed included the largest tumor diameter (in mm), tumor contour (regular/irregular), presence of acoustic shadows (yes/no), presence of solid component (yes/no), the largest diameter of the solid component (in mm), presence of papillary projections (yes/no), the number of papillary projections, the size of each papillary projection, presence of septa, the number of locules, presence of ascites and the Doppler color score (1 to 4). We also classified the lesions as unilocular, multilocular, unilocular-solid, multilocular-solid, and solid.

### 2.5. Classification Scores

#### 2.5.1. Subjective Assessment

Images were evaluated by two experienced ultrasound gynecologists with more than 15 years of experience (I.P.D. and L.A.M.) and described according to IOTA criteria, with a final classification in benign or malignant masses. In case of discrepancy, agreement ultrasound images were discussed to give a final result. Both examiners were blind to definitive histopathological diagnosis and other clinical data.

#### 2.5.2. IOTA Simple Rules (SR)

Contains 10 features to try to differentiate benign and malignant adnexal lesions. These are five features suggestive of benignity (B rules) and five features suggestive of malignancy (M rules). If the lesion shares or does not contain any characteristics of benignity and malignancy, the mass shall be considered inconclusive. These features are summarized in Table 1.

#### 2.5.3. IOTA Simple Rules Risk Assessment (SRRA)

This prediction model is available at the website: https://homes.esat.kuleuven.be/~sistawww/biomed/ssrisk/ (accessed on 1 March 2023). It uses the same parameters of SR adding, adding the detail of whether the scan was performed in an oncology center.

We used a cut-off of 10% risk of malignancy to classify the mass. If the estimated risk of malignancy was equal to or higher than 10%, the mass was considered malignant.

#### 2.5.4. The ADNEX Model

Parameters for this model were patient’s age (years), oncological center (yes/no), maximum diameter of the lesion (mm), maximum diameter of the largest solid part (mm), more than 10 locules (yes/no), number of papillae (0, 1, 2, 3, >3), presence of acoustic shadows (yes/no), presence of ascites (yes/no), serum CA125 (UI/mL) (Table 2). Results are provided as percentages for benignity and malignancy (borderline tumor, invasive cancer stage I, invasive cancer stage II-IV and metastasis to the ovary).

The referral cut-off point for malignancy was taken as ≥10%. This prediction model is available at the website: https://www.iotagroup.org/sites/default/files/adnexmodel/IOTA%20-%20ADNEX%20model.html (accessed on 1 March 2023).

#### 2.5.5. O-RADS

The lexicon used in the O-RADS system is based on IOTA descriptions, mainly type of lesion, color score, tumor size, number of papillary projections, tumor contour, as well as some IOTA benign simple descriptors, such as endometrioma, dermoid cyst, simple cyst [18]. The lesions were classified in O-RADS 0 to 5 (Table 3). We obtained six classification categories based on the probability of malignancy [19,20], as described before. 

### 2.6. Tumoral Markers

All women underwent peripheral blood sampling by venipuncture to assess CA125 levels on the same day as the ultrasound evaluation. The automated assay was performed using Alinity i CA125 II Reagent Kit (Abbot, Chicago, IL, USA).

### 2.7. Histological Diagnosis

The histological diagnosis was considered the standard reference classification of benign and malignant lesions. The extracted surgical material was analyzed by a group of pathology experts in gynecological pathology who classified the adnexal lesions in accordance with the guidelines of the World Health Organization [21]. Staging of malignant tumors according to the International Federation of Gynaecology and Obstetrics classification [22,23]. Borderline tumors are considered malignant for classification purposes in this study.

### 2.8. Statistical Analysis

Excel software for Microsoft 365 MSO (64-bit version 2211) (Redmon, WA, USA) was used for both data recording and data processing. This software package also provided the basic statistics and facilitated the calculation for comparison between variables. PSPP v1.6.2, Open Source software specialized in data analysis and equivalent to SPSS (proprietary software), was also used. For the analysis of the variables in this study, being categorical, the results are presented as a subset of numbers by class and percentage. For comparison between categorical variables, Pearson’s Chi-square test was used. The significance level was set at <0.05. For the evaluation of each model with respect to the gold standard, the different confusion matrices were made with Excel for each of the above indicators, with which specificity, sensitivity, accuracy, overall error, precision, F1-Score, positive/negative predictive value, and the corresponding Positive/Negative Likelihood Ratios can be calculated. We did not estimate sample size nor calculate statistical power.

## 3. Results

A total of 122 women were included in the present study. Eighty-one patients had a benign tumor, and forty-one patients had a malignant tumor. Histopathological definitive benign and malignant adnexal masses included are shown in Table 4.

Patients’ mean age was 51.4 years (standard deviation: 15.7; range: 14–91 years). Sixty-two (50.8%) women were postmenopausal, and fifty-four (44.3%) women were nulliparous. Most of them were asymptomatic (39.3%; *n*: 48) or had gastro-intestinal symptoms (*n*: 41; 33.6%), and only 10 patients (8.2%) had some bleeding symptoms (Table 5). Patients with malignant lesions were more frequently postmenopausal women.

At surgery, forty-one lesions depended from the right ovary (33.6%), 42 from the left ovary (34.4%), and 39 (32.0%) were bilateral. Bilateralty was not associated with malignancy. Most malignant masses were treated surgically by laparotomy (29 out of 41, 70.7%), while most benign ones had a laparoscopic approach (59 out of 81; 72.8%). Surgical procedures included unilateral cystectomy, oophorectomy, or salpingo-oophorectomy in 60 patients and bilateral salpingo-oophorectomy in 62 patients. Hysterectomy was performed in 32 patients (26.2%). Both hysterectomy and bilateral salpingo-oophorectomy were more frequently performed in women with malignant lesions (Table 6).

Mean CA125 levels were 354.5 IU/mL (standard deviation: 1538.8 IU/mL; range: from 4.6 IU/mL to 12059.0 IU/mL). Mean CA-25 values were significantly higher in malignant lesions (892.3 IU/mL, standard deviation: 2486.5; range: from 8.6 IU/mL to 12059.0 IU/mL) as compared to benign lesions (50.5 IU/mL, standard deviation: 106.1; range: from 4.6 IU/mL to 818 IU/mL).

Regarding ultrasound data, the maximum diameter of adnexal lesions ranged from 21.0 to 289.0 mm (mean: 94.2 mm; standard deviation: 52.1 mm), most of them with a regular contour. Acoustic shadows were present in half of them (54.9%, *n*: 67). Up to 49.2% of adnexal masses had solid components, with a mean size of 51.6 mm (standard deviation: 36.5 mm; range: from 10.0 mm to 210.0 mm). Solid components showed a Doppler score of 3 or 4 in 50% of the cases. A few proportions of masses had either thick or irregular septum (*n*: 17, 13.9%) or thin septum (*n*: 27, 22.1%). Eighty-two masses were unilocular (*n*: 82, 67.2%), and only 9 had more than 10 locules (7.4%). Twenty-eight lesions included one or more papillae (*n*: 28, 22.9%), with a medium size of 25.1 mm (standard deviation: 18.7 mm; range: from 3.3 mm to 90 mm), and 35.7% of them (*n*: 10) had score color 3–4. Ascites were present in 12 cases (9.8%) (Table 7).

Sensitivity, specificity, positive and negative predictive value, Odds Ratio and Area under the curve of Subjective Assessment, Simple Rules, Simple Rules Risk Assessment, ADNEX model with or without CA125, and ORADS are shown in Table 8. The highest sensitivity was for ADNEX Model with CA125 (95.1% (88.7–100)) superior to ORADS (90.2% (81.6–98.8)), Subjective assessment (87.8% (78.4–97.2)), ADNEX Model without CA125 (87.8% (78.4–97.2)), Simple Rules Risk Assessment (78.1% (66.9–89.3)) and Simple Rules (66.7% (52.2–81.2)). The highest specificity was for Simple Rules (89.2% (82.1–96.3)), ADNEX Model with CA125 (74.1% (65.9–82.3)), Simple Rules Risk Assessment (72.8% (64.5–81.1)), Subjective assessment (69.1% (60.7–77.5)), ADNEX Model without CA125 (67.9% (59.5–76.3)) and O-RADS (60.5% (52.2–68.8)). Positive predictive value (PPV) was better for Simple Rules (72.0%), and negative predictive value (NPV) was superior to 86% in all cases. Odds Ratio (OR) ranged from 55.7 (ADNEX Model with CA125) to 9.5 (SSRA).

The Area under the Curve (AUC) was better in ADNEX Model with CA125 (0.88) than in ADNEX Model without CA125 (0.84), subjective assessment (0.81), Simple Rules Risk Assessment (0.80), Simple Rules (0.78) and O-RADS (0.75) (Figure 1)

## 4. Discussion

### 4.1. Summary of Findings

In the present study, we analyzed how IOTA SR, IOTA SRRA, the ADNEX model with or without CA125, and O-RADS can help in the discrimination between benign and malignant adnexal masses in the same set of patients.

The mean patient age of the sample obtained was 51.4 years, without significant differences between the lesions observed between the benignity and malignancy groups. Other studies have also found that the majority of patients with ovarian cancer are between 50–59 years old [24]. Our series includes only one patient in each group of 14 years old (one malignant dysgerminoma and one cystic teratoma), and the rest are all adults older than 23 years old.

Regarding menopausal status, in our study, malignant masses were more frequent in postmenopausal women (65.9%) than premenopausal (43.2%) (*p* = 0.018), according to the literature [25]. 

Other parameters, such as parity (nulliparous vs. parous) or body mass index (BMI), do not show differences between both groups. However, of the 17 patients with advanced ovarian cancer (stages II–IV), only 3 were premenopausal (17.6%).

Adnexal masses, whether benign or malignant, may result from imaging findings since they remain asymptomatic, even in advanced stages. In our series, 39.3% (*n*: 48) of patients were asymptomatic (6 of them in stage II-IV of ovarian cancer), 33.6% (*n*: 41) had digestive symptoms such as pelvic pain, abdominal swelling or discomfort, or gynecological bleeding (8.2%, *n*: 10) which are nonspecific symptoms of ovarian cancer [5]. In other series, reasons for the initial ultrasound examination were pelvic pain, menstrual disorder, abdominal swelling or discomfort, referral for a second opinion, and abnormal finding during a routine gynecological check-up [26].

The laparoscopic approach was more frequent in the management of benign masses (benign: 72.8% vs. malignant: 29.3%). This is in agreement with the literature, which estimates that in the removal of suspicious ovarian masses by laparoscopy, there is a certain risk of rupture and, therefore, of the spread of the disease, even in the early stages of ovarian cancer [27,28]. Hysterectomy was more commonly performed in malignant lesions, as expected. 

There were no differences in malignancy between right or left adnexal masses (total right: 33.6%, left: 34.4%). Thirty-nine patients (32.0%) had bilateral lesions. Fourteen of them had malignant involvement (eight bilateral serous carcinomas, one bilateral borderline carcinoma, one bilateral clear cell carcinoma, one bilateral endometrioid carcinoma, one serous carcinoma, and cystoadenofibroma, one borderline and endometrioma and teratoma, and one borderline and serous cystoadenoma).

The mean value of CA125 was 354.5 IU/mL, which was significantly lower in benign lesions (50.5 IU/mL) than in malignant tumors (892.3 U/mL). According to a recent review high level of CA125 (≥30 IU/mL) has a sensitivity of 81% and specificity of 75% for distinguishing benign from malignant tumors in mixed pre and postmenopausal women with adnexal masses. However, it has low sensitivity (50%) to distinguish the early stage of ovarian cancer [29]. In our series mean CA125 in early stages of ovarian cancer (including borderline and stage I) was 240.0 IU/mL with a wide range of values (from 8.6 IU/mL to 1579.3 IU/mL). Additionally, in premenopausal women, it has low specificity. In our series, premenopausal women with ovarian cancer had high levels of CA125 (mean: 238.7 UI/mL), but only 8/14 women (57.1%) had levels higher than 30 IU/mL. Out of the seven cases of clear cell carcinoma, five cases had levels ≤ 50 UI/mL. Xie et al. have shown that the combination of IOTA Simple Rules or O-RADS CA125 may improve the ability to distinguish benign from malignant ovarian tumors, and the AUCs of IOTA SR combined with CA125, O-RADS combined with CA125, and IOTA SR plus O-RADS combined with CA125 were 0.900, 0.891, and 0.909, respectively [29]. Additionally, it has recently been published that the study of IOTA SR or O-RADS in combination with CA125 may improve the ability to distinguish benign from malignant ovarian tumors [30].

The sensitivity of subjective assessment in our study to detect malignant masses is 87.8% (78.4–97.2) which is similar to other studies in which the sensitivity of this approach ranged from 56.4% to 100% [7,9]. However, the specificity (69.1%) is lower than expected according to the literature (84.2–99.9%) [7,9]. This is probably because of a selection bias in our study since many of the adnexal lesions included had been classified as suspicious and required histological study, as evidenced by the high percentage of malignancy according to the pathological study (33.6%). While NPV (91.8%) is maintained at appropriate levels, PPV of 59.0% is relatively low.

As stated above, IOTA simple rules include five benignity characteristics (B features) and five malignancy characteristics (M features). When there are characteristics of both benignity and malignancy or no characteristic of malignancy or benignity appears, the diagnosis of the mass is considered inconclusive, which in our study reached 24.6% (*n*: 30). Qian et al. compared the diagnostic performance of different ultrasound models for adnexal masses among which was IOTA SR [31]. They defined two groups depending on whether inconclusive masses were considered malignant (Group 1) or benign (Group 2), observing a decrease in sensitivity (Group 2: 69%) and an increase in specificity (Group 2: 96%), quite similar to our findings. In our experience, although the specificity obtained in our study for IOTA SR (89.2%) is similar to that reflected in other studies (71.7–98.6) [32,33], sensitivity is lower, according to that obtained from Qian et al. [31].

For IOTA simple rules risk assessment (SRRA), we considered 10% of the risk of malignancy as the cut-off point for malignancy. Sensitivity (78.1%) is lower than what was found by Hiett et al. (100%), while specificity in our study was higher (72.8% versus 51.8%) [33]. The AUC remains similar, which would determine that SRRA is a good method for classifying adnexal masses.

ADNEX Model with and without CA125 can distinguish benign and malignant lesions, providing a percentage of probability of benignity, malignancy, borderline carcinoma, and even of staging within ovarian tumors (stage I, II-IV, metastatic) with an AUC of 0.88 (ADNEX with CA125) or 0.84 (ADNEX without CA125), proving to be the best method of ultrasound discrimination between benign and malignant masses in this study. The cut-off point to distinguish between malignant or benign masses has been considered at 10%. Jeong et al. studied the diagnostic performance of different cut-off points of overall malignancy risks observing that as the cut-off point increased (5–10–15%), the specificity increased, maintaining the sensitivity at 0.90 [8]. They estimated that the optimal cut-off point could be 47.3%, reaching a specificity of 0.98. In our sample, adding CA125 to the ADNEX model improved sensitivity, specificity, and positive and negative predictive values. Chen et al. found similar sensitivity and specificity in the ADNEX model with and without CA125 (91.4% vs. 91.4%; 78.9% vs. 79.5%) [34]. Peng et al. also studied the sensitivity for diagnosing borderline, stage I, and metastatic ovarian tumors, and they found that it was only 60.0%, 28.6%, and 45.5%, respectively [35]. In our series, the mean percentage of probability of being a borderline tumor or stage I in the ADNEX model with/without CA125 is over 10%, while stage II-IV rises up to 53.0%. Other studies have also directly studied the concordance between ultrasound and laparoscopy in the assessment of the intrabdominal tumor spread [36].

O-RADS is the last ultrasound score introduced for classifying adnexal masses in benign or malignant groups, including suggestions for management according to the risk category. It also includes lexicon descriptors of the most frequent benign adnexal masses (O-RADS score 2). O-RADS scores 4 and 5 correlate with an intermediate (10–49%) or high (≥50%) risk of malignancy and are usually considered as the cut-off point. Our results in terms of sensitivity and specificity (90.2% and 60.5%, respectively) are in agreement with other studies that have been published. The reported sensitivity varies from 72% to 100%, and specificity from 36.9% to 99.6% [19,20,30,32,33,34,37,38,39,40]. Moreover, Cao et al. studied including O-RADS 3 as the limit to consider malignancy rising sensitivity from 87.7% to 98.7%, at the expense of diminishing specificity (98.7% versus 83.7%) [20]. 

There are few recent studies that compare different ultrasound scores in order to differentiate benign or malignant masses. Xie et al., assessed the efficacy of IOTA simple rules, O-RADS, and CA125 to distinguish benign and malignant adnexal masses [30]. They concluded that a combination of IOTA SR or O-RADS in combination with CA125 may improve the ability to distinguish benign from malignant ovarian tumors. Lai et al., compared O-RADS, GI-RADS, and ADNEX models as an external validation study conducted by junior sonologists demonstrating that the three methods worked correctly and can be selected according to the type of center, access to patients’ clinical data or personal comfort [40]. Chen et al. analyzed the performance of the O-RADS and ADNEX models in 322 patients and concluded that both were comparable and the O-RADS had higher sensitivity than the ADNEX model (96.6% vs. 91.4%) and relatively similar specificity [34]. Jeong et al. compared the ADNEX model with the subjective assessment of gynecologic experts in differentiating ovarian diseases and found that both methods were equal and that ADNEX may help gynecologic beginners [8]. Basha et al. compared O-RADS, GI-RADS, and IOTA simple rules stating that O-RADS had higher sensitivity than GI-RADS and IOTA simple rules with relatively similar specificity and reliability [32]. Guo et al. compared O-RADS, IOTA SR, and other ultrasound systems by senior and junior doctors concluding that O-RADS performed best and showed the highest sensitivity [38]. Hiett et al. studied the performance of IOTA Simple Rules, Simple Rules risk assessment, ADNEX model, and O-RADS in North American women concluding that all are useful, but the IOTA models have higher specificity [33].

### 4.2. Limitations

This study reflected the comparison of different ultrasound scores in classifying adnexal masses. However, its generalization may present limitations given the reduced sample, that bilateral lesions were included taking into account the biggest or the worse prognosis, which could interfere with tumor markers (CA125) used for the calculation of the ADNEX model. Additionally, the classification of adnexal masses was made by experts in gynecological ultrasound, which could have disrupted the results given that most of the adnexal lesions are first detected by inexperienced sonographers. Our study was a retrospective study but blind to histopathology, which has tried to remedy the clinical situation in which the sonographer is evaluating the adnexal mass. In addition, only those cases in which the images were suitable for evaluation were selected, which may change results in situations in which, due to obesity, gas interposition, or poor transmission, the images are not always easy to classify. Finally, the fact that this study was performed in only one center has the advantage that the classification criteria are more homogeneous.

### 4.3. Future Research Directions

At the moment, there is no perfect ultrasound score that differentiates between malignant and benign adnexal masses. It would be interesting to study whether the combination of several scores could give uniform criteria for classification and management.

## 5. Conclusions

The use of different ultrasound scores (IOTA SR, IOTA SRRA, ADNEX model with or without CA125, and O-RADS) can be used in the differentiation of benign and malignant masses and are similar to the subjective assessment of an experienced sonographer.

## Figures and Tables

**Figure 1 diagnostics-13-01307-f001:**
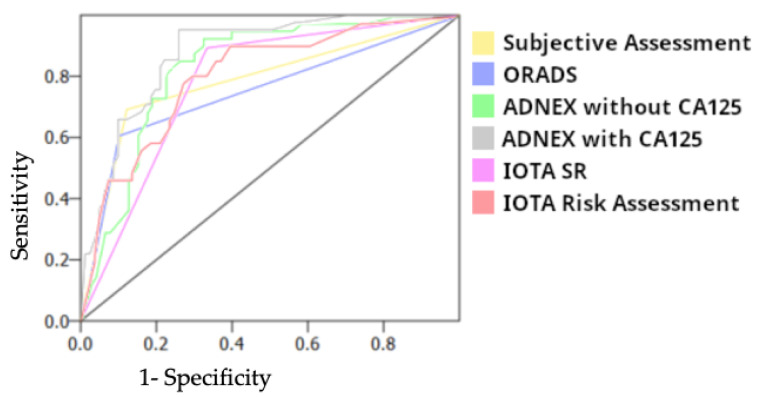
Area under the curve for different approaches.

**Table 1 diagnostics-13-01307-t001:** IOTA Simple rules.

Benignity Characteristics (B Features)
B1: Unilocular B2: Presence of solid component < 7 mm maximum diameter B3: Acoustic shadows B4: Regular multilocular tumor with largest diameter < 100 mm B5: No blood flow (score color 1)
Malignancy characteristics (M features)
M1: Irregular solid tumor M2: Ascites M3: At least 4 papillary structures M4: Multilocular irregular solid tumor with maximum diameter ≥ 100 mm M5: Very strong blood flow (score color 4)

**Table 2 diagnostics-13-01307-t002:** Variables assessed in the ADNEX model.

Variable	Description
Age	Years
Oncology center	Yes/no
Maximal diameter of the lesion	Expressed in mm
Maximal diameter of the solid part	Expressed in mm
>10 locules	Yes/No
Number of papillae	0/1/2/3/>3
Acoustic shadow	Yes/No
Ascites	Yes/No
CA125	IU/mL

**Table 3 diagnostics-13-01307-t003:** Ovarian-Adnexal Reporting and Data System classification (O-RADS).

O-RADS	Probability of Malignancy	Description
0		Incomplete assessment
1	0%	Normal/Functional
2	<1%	Most probably normal
3	1–9%	Low risk of malignancy
4	10–49%	Intermediate risk of malignancy
5	≥50%	High risk of malignancy

**Table 4 diagnostics-13-01307-t004:** Histologic diagnosis of benign lesions.

	**Total**	**Premenopausal** **(*n*)**	**Postmenopausal** **(*n*)**	
Histology for benign lesions	81	46	35	
Dermoid cyst	19	15	4	
Endometrioma	12	11	1	
Fibroma	11	3	8	
Serous cystadenoma	11	4	7	
Mucinous cistoadenoma	8	5	3	
Cistoadenofibroma	6	1	5	
Brenner tumor	3	0	3	
Tubo-ovarian abscess	3	2	1	
Functional cyst	2	2	0	
Hidrosalpinx	2	2	0	
Paraovarian cyst	2	1	1	
Hiperthecosis	1	0	1	
Fibrotecoma	1	0	1	
	**Total**	**Premenopausal** **(*n*)**	**Postmenopausal** **(*n*)**	**Stage** **(*n*)**
Histology for malignant lesions	41	14	27	
Ovarian serous carcinoma	15	2	13	2 IA, 1 IC, 1 II, 2 IIIA, 5 IIIC
Clear cell carcinoma	7	2	5	4 IC, 1 IIA, 1 IIB, 1 IIIA
Serous borderline carcinoma	6	5	1	6 IA
Mucinous borderline carcinoma	3	1	2	3 IA
Endometrioid carcinoma	3	1	2	2 IA, 1 IIIC
Tubarian serous carcinoma	1	0	1	III C
Neuroendocrine carcinoma	1	1	0	
Stromal hyperplasia	1	0	1	
Steroid cell carcinoma	1	0	1	
Disgerminoma	1	1	0	IC
Struma ovarii	1	1	0	IA
Sex cord tumor	1	0	1	IA

**Table 5 diagnostics-13-01307-t005:** Baseline characteristics of the patients in the present study.

Baseline Conditions	Total (*n*: 122)	Benign (*n*: 81)	Malignant (*n*: 41)	*p* Value
Age (years)	51.4 (SD: 15.7; range:14–91)	49.8 (SD: 15.7; range: 14–81)	54.8 (SD: 15.2; range: 14–91)	0.100
Menopause				0.018
Yes	62 (50.8%)	35 (43.2%)	27 (65.9%)	
No	60 (49.2%)	46 (56.8%)	14 (34.1%)	
Parity				0.137
Nulliparous	54 (44.3%)	32 (39.5%)	22 (53.7%)	
Parous	68 (55.7%)	49 (60.5%)	19 (46.3%)	
BMI (kg/m)	26.6 (SD: 5.4; range 18.6–42.0)	28.0 (SD: 5.5; range: 18.8–42.0)	26.6 (SD: 6.0; range: 18.6–41.0)	0.409
Symptoms				0.362
Asymptomatic	48 (39.3%)	36 (44.4%)	12 (29.3%)	
Digestive	41 (33.6%)	25 (30.9%)	16 (39.0%)	
Bleeding	10 (8.2%)	7 (8.6%)	3 (7.3%)	
Other	23 (18.9%)	13 (16.0%)	10 (24.4%)	

SD: standard deviation.

**Table 6 diagnostics-13-01307-t006:** Surgical procedures and surgical findings.

Surgery	Total (*n*: 122)	Benign (*n*: 81)	Malignant (*n*: 41)	*p* Value
Surgical approach				<0.001
Laparoscopic	71 (58.2%)	59 (72.8%)	12 (29.3%)	
Laparotomy	51 (41.8%)	22 (27.2%)	29 (70.7%)	
Surgical procedure				
Hysterectomy	32 (26.2%)	9 (11.1%)	23 (56.1%)	<0.001
Unilateral adnexal surgery	60 (49.2%)	50 (61.7%)	10 (24.4%)	<0.001
Bilateral salpingo-oophorectomy	62 (50.8%)	31 (38.3%)	31 (75.6%)	<0.001
Laterality of the lesion at surgery				0.512
Right	41 (33.6%)	30 (37.0%)	11 (26.8%)	
Left	42 (34.4%)	27 (33.3%)	15 (36.6%)	
Bilateral	39 (32.0%)	24 (29.6%)	15 (36.6%)	

**Table 7 diagnostics-13-01307-t007:** Ultrasound findings in this series.

Ultrasound Features	Total (*n*: 122)	Benign (*n*: 81)	Malignant (*n*: 41)	*p* Value
Largest size (mm)	94.2 (SD: 52.1; range: 21.0–289.0)	90.2 (SD: 53.3; range: 23.0–289.0)	102.2 (SD: 49.3; range: 21.0–210.0)	0.233
Contour:				<0.001
Regular	99 (81.1%)	76 (93.8%)	23 (56.1%)	
Irregular	23 (18.9 %)	5 (6.2%)	18 (43.9%)	
Acoustic shadow:				<0.001
Yes	67 (54.9%)	56 (69.1%)	11 (26.8%)	
No	55 (45.1%)	25 (30.9%)	30 (73.2%)	
Presence of solid areas:				<0.001
Yes	60 (49.2%)	25 (30.9%)	35 (85.4%)	
No	62 (50.8%)	56 (69.1%)	6 (14.6%)	
Size of solid areas (mm)	51.6 (SD: 36.5; range: 10.0–210.0)	49.6 (SD: 33.5; range: 10.0–112.0)	53.1 (SD: 39.0; range: 12–210.0)	0.721
Doppler within solid areas (Score color)				<0.001
1–2	30 (50.0%)	19 (76.0%)	11 (31.4%)	
3–4	30 (50.0%)	6 (24.0%)	24 (68.6%)	
Septum:				0.351
None	78 (63.9%)	52 (64.2)	26 (63.4%)	
Thin	27 (22.1%)	20 (24.7%)	7 (17.1%)	
Thick/Irregular	17 (13.9%)	9 (11.1%)	8 (19.5%)	
Doppler within septum (Score color)				0.02
1–2	30 (68.2%)	24 (85.7%)	6 (37.5%)	
3–4	14 (31.8%)	4 (14.3%)	10 (62.5%)	
Number of locules				0.591
0 (solid mass)	4 (3.3%)	2 (2.5%)	2 (4.9%)	
1	78 (63.9%)	55 (67.9%)	23 (56.1%)	
2–9	31 (25.4%)	19 (23.5%)	12 (29.3%)	
≥10	9 (7.4%)	5 (6.2%)	4 (9.8%)	
Number of papillae				0.011
0	94 (77.0%)	69 (85.2%)	25 (61.0%)	
1	11 (9.0%)	5 (6.2%)	6 (14.6%)	
>1	17 (13.9%)	7 (8.6%)	10 (24.4%)	
Size papillae (mm):	25.1 (SD: 18.7; range: 3.3–90)	12.0 (SD: 3.71; range: 4.0–90)	34.1(SD: 20.3; range: 3.3–47)	0.002
Doppler within papillae (Score color)				0.114
1–2	18 (64.3%)	10 (83.3%)	8 (50.0%)	
3–4	10 (35.7%)	2 (16.7%)	8 (50.0%)	
Ascites:				0.103
No-mild	110 (90.2%)	76 (93.8%)	34 (82.9%)	
Moderate-Severe	12 (9.8%)	5 (6.2%)	7 (17.1%)	

SD: standard deviation.

**Table 8 diagnostics-13-01307-t008:** Diagnostic performance of different approaches assessed in the present study.

	Sensitivity (%)	Specificity (%)	PPV (%)	NPV (%)	OR
Subjective assessment	87.8 (78.4–97.2)	69.1 (60.7–77.5)	59.0	91.8	16.1
Simple Rules	66.7 (52.2–81.2)	89.2 (82.1–96.3)	72.0	86.6	16.6
Simple Rules Risk Assessment	78.1 (66.9–89.3)	72.8 (64.5–81.1)	59.3	86.8	9.5
ADNEX model with CA125	95.1 (88.7–100)	74.1 (65.9–82.3)	65.0	96.8	55.7
ADNEX model without CA125	87.8 (78.4–97.2)	67.9 (59.5–76.3)	58.1	91.7	15.3
O-RADS	90.2 (81.6–98.8)	60.5 (52.2–68.8)	53.6	92.5	14.2

## Data Availability

Data are available upon reasonable request.

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
