# Peer review of "Comparison of Ultrasound Scores in Differentiating between Benign and Malignant Adnexal Masses"

_diagnostics, 2023, doi:10.3390/diagnostics13071307_

Round 1

Reviewer 1 Report

Review Report

·     In this paper, the authors aimed to compare the diagnostic performance of some of these scores in the evaluation of adnexal masses in the same set of patients.

·     Based on a prospective analysis of 122 women with adnexa masses, the authors concluded that IOTA SR, IOTA SRRA, ADNEX model with or without Ca125 and O-RADS could help in the differentiation of benign and malignant masses, and their performance is similar to subjective assessment of an experienced sonographer.

·     The paper is interesting. The authors have clearly worked hard to detail their study, but I have some comments:

POINTS OF STRENGTH

1.     Interesting topic.

2.     The results are Ok.

POINTS OF WEAKNESS

1.     Lack of novelty.

2.     Small sample size.

3.     Retrospective analysis.

4.     Static images.

SPECIFIC COMMENTS

1. Introduction: Several paragraphs are without references, e.g., “It has the advantage of being a harmless technique in which the patient does not receive radiation, is relatively economically affordable compared to other imaging techniques, such as computed tomography (CT) and magnetic resonance imaging (MRI), and available to many gynecologists and radiologists world-wide.  What was the power of sample size calculation?”.

2. A flowchart of the study and the number of excluded patients are required.

3. Page 3, line 132:  “Initially, all women underwent a transabdominal abdominal ultrasound for measurement of the tumor, especially in large masses” …Please delete “abdominal”.

4. What were the years of experience of examiners who perform US examinations?

5. Why did you not try to perform an interviewer agreement to strengthen your results?

6. What version of O-RADS was used in the study, O-RADS 2018 or 2022?

7. Why do you consider borderline tumor as malignant ones?

8. Tables 4 and 5 can be merged into one Table.

9. Start the discussion with the main results of the study and avoid repeating words.

10. The grammar and language of the manuscript need further polishing.

Author Response

In this paper, the authors aimed to compare the diagnostic performance of some of these scores in the evaluation of adnexal masses in the same set of patients.

  • Based on a prospective analysis of 122 women with adnexa masses, the authors concluded that IOTA SR, IOTA SRRA, ADNEX model with or without Ca125 and O-RADS could help in the differentiation of benign and malignant masses, and their performance is similar to subjective assessment of an experienced sonographer.
  • The paper is interesting. The authors have clearly worked hard to detail their study, but I have some comments:

POINTS OF STRENGTH

  1. Interesting topic.
  2. The results are Ok.

POINTS OF WEAKNESS

  1. Lack of novelty.
  2. Small sample size.
  3. Retrospective analysis.
  4. Static images.

SPECIFIC COMMENTS

Comment Introduction: Several paragraphs are without references, e.g., “It has the advantage of being a harmless technique in which the patient does not receive radiation, is relatively economically affordable compared to other imaging techniques, such as computed tomography (CT) and magnetic resonance imaging (MRI), and available to many gynecologists and radiologists world-wide. 

Answer: We added a new reference (Khiewvan B, Torigian DA, Emamzadehfard S, Paydary K, Salavati A, Houshmand S, Werner TJ, Alavi A. An update on the role of PET/CT and PET/MRI in ovarian cancer. Eur J Nucl Med Mol Imaging. 2017 Jun;44(6):1079-1091) (changed in paper)

Comment What was the power of sample size calculation?

Answer: We did not estimate sample size nor calculate statistical power.

Comment: A flowchart of the study and the number of excluded patients are required.

Answer: we added the following sentence: We performed 186 surgeries related to suspected adnexal masses. Two of them were excluded because they were finally diagnosed of cervical schwanoma and pediculated myoma. Other sixty-two lesions were studied by non-expert sonographers and were not included in the present study. (changed in paper)

Comment: Page 3, line 132:  “Initially, all women underwent a transabdominal abdominal ultrasound for measurement of the tumor, especially in large masses” …Please delete “abdominal”.

Answer: Done as suggested (changed in paper)

Comment: What were the years of experience of examiners who perform US examinations?

Answer: This information is added. Ultrasound examinatios were performed by experienced gynecologists with more than 10 years of experience (added in paper).

Comment: Why did you not try to perform an interviewer agreement to strengthen your results?

Answer: It wasn´t the objective of the study. We akn.

Comment. What version of O-RADS was used in the study, O-RADS 2018 or 2022?

Answer: We used the 2018 version, as described in the Methodology section we used O-RADS 2018 (Andreotti RF, Timmerman D, Benacerraf BR, Bennett GL, Bourne T, Brown DL, et al. Ovarian-Adnexal Reporting Lexicon for Ultrasound: A White Paper of the ACR Ovarian-Adnexal Reporting and Data System Committee. J Am Coll Radiol. 2018;15:1415-29)

Comment. Why do you consider borderline tumor as malignant ones?

Answer: Borderline ovarian tumors (BOT) are neoplasms of epithelial origin characterized by up-regulated cellular proliferation and the presence of slight nuclear atypia but without destructive stromal invasion. BOT can be associated with microinvasion, intraepithelial carcinoma, lymph node involvement, and non-invasive peritoneal implants. The vast majority of BOT are limited to the ovary(ies) at presentation with 75% being diagnosed at FIGO stage I, compared to only 10% of ovarian carcinomas diagnosed at an early stage. They generally have an excellent prognosis with a 10-year survival of 97% for all stages combined, although recurrences and malignant transformation can occur. (Taken from Hauptmann S, Friedrich K, Redline R, Avril S. Ovarian borderline tumors in the 2014 WHO classification: evolving concepts and diagnostic criteria. Virchows Arch. 2017 Feb;470(2):125-142. doi: 10.1007/s00428-016-2040-8. Epub 2016 Dec 27. PMID: 28025670; PMCID: PMC5298321.) (not included explanation in paper)

Comment. Tables 4 and 5 can be merged into one Table.

Answer: Done (changed in paper, and changed the numbering of successive tables and their references in the text)

Comment. Start the discussion with the main results of the study and avoid repeating words.

Answer: Done as suggested. (changed in paper).

Comment. The grammar and language of the manuscript need further polishing.

Answer: We have tried to improve language.

All amendments for reviewer 1 highlighted in red

Reviewer 2 Report

1.     Please check and correct the whole manuscript. 

Examples

Line 95: “…(O-RADS) [15] It…” to “…(O-RADS) [15]. It…”.

Line 127: “…and other) other),…” to “…and other),…”

2.     Even in non-Caucasian postmenopausal women, “there is a more proportion” of malignant masses, at least for epithelial ovarian cancers etc. Thus, please comment on your manuscript “Regarding menopausal status, there is a SLIGHT more proportion of malignant masses in menopausal women”. 

Ref: Shen F, et al. The prevalence of malignant and borderline ovarian cancer in pre- and post-menopausal Chinese women. Oncotarget 2017, 8:80589-94. 

https://pubmed.ncbi.nlm.nih.gov/29113327/

3.     Some important references related to your article are missing. 

Examples 

Iatrakis GM, et al. A new risk malignancy index to predict ovarian

cancer: a bicentric preliminary study. J BUON 2018, 23:1380-3. https://pubmed.ncbi.nlm.nih.gov/30570861/

Moruzzi MC, et al. Diagnostic performance of ultrasound in assessing the extension of disease in advanced ovarian cancer. Am J Obstet Gynecol 2022, 227:601.e1-601.e20. https://pubmed.ncbi.nlm.nih.gov/35752305/

Author Response

Comment: Please check and correct the whole manuscript. Examples Line 95: “…(O-RADS) [15] It…” to “…(O-RADS) [15]. It…”.Line 127: “…and other) other),…” to “…and other),…”

Answer: We have reviewed the manuscript. Corrections done

Comment: Even in non-Caucasian postmenopausal women, “there is a more proportion” of malignant masses, at least for epithelial ovarian cancers etc. Thus, please comment on your manuscript “Regarding menopausal status, there is a SLIGHT more proportion of malignant masses in menopausal women”. Ref: Shen F, et al. The prevalence of malignant and borderline ovarian cancer in pre- and post-menopausal Chinese women. Oncotarget 2017, 8:80589-94. 

https://pubmed.ncbi.nlm.nih.gov/29113327/

Asnwer: We have modified the manuscript and added the new reference.

            Comment:  Some important references related to your article are missing. 

Answer: references added as suggested. Except that from Iatrakis GM, et al. A new risk malignancy index to predict ovarian cancer: a bicentric preliminary study. J BUON 2018, 23:1380-3. We think this paper is not very related to our study.

All amendments for reviewer 2 are highlighted in blue

Reviewer 3 Report

The manuscript submitted for review concerns an interesting and important topic of using various ultrasound methods to predict the malignancy risk of ovarian tumors.

The manuscript is well written and covers a wide range of methods studied and their expert evaluation.

My minor remarks:

1. There is no consistency in the use of symbols for CA 125 (different spellings appear: Ca125 Ca-125 CA-125 CA 125 etc.) This requires unification

2. In what situations was a hysterectomy without BSO performed in patients with ovarian tumors? (l129)

3. It is not made clear that the abbreviation IPD/LAM refers to the authors of the manuscript (l158)

4. Minor misspellings (l178 - proyections, l180 - invasica, l364 - founf, l365 - was was)

5. The authors in many places write about "menopausal women" (e.g. l226). In my opinion, it should be "postmenopausal"

6. Error in the number of postmenopausal patients in Table 5. It should be 27 instead of 25

7. Below table 7 unnecessary explanation of the abbreviation BMI.

8. Table 8 - the first two lines are the same.

9. Figure 1 - hard to read as black and white. It will be better published in color.

10. Some parts of the discussion present the results, not their confrontation with the literature.

11. Incorrectly citing sensitivity and specificity as 0.90% and 0.98% (l359-360)

12. Error in conclusions, IOTA SRR rather IOTA SRRA.

After these minor corrections, the manuscript will be ready for publication.

Author Response

The manuscript submitted for review concerns an interesting and important topic of using various ultrasound methods to predict the malignancy risk of ovarian tumors. The manuscript is well written and covers a wide range of methods studied and their expert evaluation. My minor remarks:

Comment: There is no consistency in the use of symbols for CA 125 (different spellings appear: Ca125 Ca-125 CA-125 CA 125 etc.) This requires unification

Answer: Done as suggested (changed in paper) CA125

Comment: In what situations was a hysterectomy without BSO performed in patients with ovarian tumors? (l129)

Answer: There is a mistake in the expression: it should be “… surgical procedure (laparoscopic/laparotomy,  uni or bilateral adnexal surgery with or without hysterectomy), and ….” (changed in paper)

Comment: It is not made clear that the abbreviation IPD/LAM refers to the authors of the manuscript (l158)

Answer: Yes: IPD: Irene Pelayo Delgado; LAM: Leopoldo Abarca Martínez)  (changed in paper)

Comment: Minor misspellings (l178 - proyections, l180 - invasica, l364 - founf, l365 - was was) Answer: Sorry. We have made amendments (changed in paper)

Comments. The authors in many places write about "menopausal women" (e.g. l226). In my opinion, it should be "postmenopausal"

Answer: Modified as suggested changed in paper)

Comment: Error in the number of postmenopausal patients in Table 5. It should be 27 instead of 25

Answer: Corrected (changed in paper)

Comment: Below table 7 unnecessary explanation of the abbreviation BMI.

Answer. Amended (changed in paper)

Comment: Table 8 - the first two lines are the same.

Answer: Amended (changed in paper)

Comment: Figure 1 - hard to read as black and white. It will be better published in color.

Answer: We are sorry you can only read it in black and white, in the version of the paper that I handle it´s already in colours.

Comment: Some parts of the discussion present the results, not their confrontation with the literature.

Answer; Reviewer is right. We have impreoved this. New reference addes )Salvador S, Scott S, Glanc P, Eiriksson L, Jang JH, Sebastianelli A, Dean E. Guideline No. 403: Initial Investigation and Management of Adnexal Masses. J Obstet Gynaecol Can. 2020 Aug;42(8):1021-1029.e3 (added in paper). Biggs WS, Marks ST. Diagnosis and Management of Adnexal Masses. Am Fam Physician. 2016 Apr 15;93(8):676-81 (added in paper)

Comment: Incorrectly citing sensitivity and specificity as 0.90% and 0.98% (l359-360)

Answer: Sorry for this mistake. Amended(changed in paper)

Comment: Error in conclusions, IOTA SRR rather IOTA SRRA.

Answer: Reviewer is right. Amended (changed in paper)

All amendments for reviewer 3 are highlighted in green

Round 2

Reviewer 1 Report

The authors have performed a good job and responded to all reviewers' comments